# Ultracold plasmas from strongly anti-correlated Rydberg gases in the Kinetic Field Theory formalism

Elena Kozlikin[1][*], Robert Lilow[2], Martin Pauly[1], Alexander Schuckert[3], Andre Salzinger[4], Matthias Bartelmann[1] and Matthias Weidemüller[4]

**1** Heidelberg University, Institute for Theoretical Physics, Heidelberg, Germany
**2** Department of Physics, Technion, Haifa, Israel
**3** Joint Quantum Institute and Joint Center for Quantum Information and Computer Science, NIST and University of Maryland, College Park, MD 20742, USA
**4** Heidelberg University, Physikalisches Institut, Heidelberg, Germany
[*] elena.kozlikin@uni-heidelberg.de

## Abstract

The dynamics of correlated systems is relevant in many fields ranging from cosmology to plasma physics. However, they are challenging to predict and understand even for classical systems due to the typically large numbers of particles involved. Here, we study the evolution of an ultracold, correlated many-body system with repulsive interactions and initial correlations set by the Rydberg blockade using the analytical framework of Kinetic Field Theory (KFT). The KFT formalism is based on the path-integral formulation for classical mechanics and was first developed and successfully used in cosmology to describe structure formation in Dark Matter. The theoretical framework offers a high flexibility regarding the initial configuration and interactions between particles and, in addition, is computationally cheap. More importantly, the analytic approach allows us to gain better insight into the processes which dominate the dynamics. In this work we show that KFT can be applied in a much more general context and study the evolution of a correlated ion plasma. We find good agreement between the analytical KFT results for the evolution of the correlation function and results obtained from numerical simulations. We use the correlation functions obtained with KFT to compute the temperature increase in the ionic system due to disorder-induced heating. For certain choices of parameters we observe that the effect can be reversed, leading to correlation cooling. Due to its numerical efficiency as compared to numerical simulations, a detailed study using KFT can help to constrain parameter spaces where disorder-induced heating is minimal in order to reach the regime of strong coupling.

# 1 Introduction

The study of ultracold plasmas opens up the possibility to access the regime of strong coupling where the Coulomb interaction energy exceeds the kinetic energy of the ions [1,2]. The Coulomb coupling parameter,

$$\Gamma = e^2 \frac{\sqrt[3]{\frac{4\pi n_{\mathrm{i}}}{3}}}{k_{\mathrm{B}} T_{\mathrm{i}}} \,, \tag{1}$$

can reach values of the order of $\Gamma \approx 10^3-10^6$ in this regime. Strong coupling can give rise to many-body effects and strong spatial correlations between particles. Ultracold plasmas thus present an ideal laboratory to study the properties of exotic phases of matter such as laser-induced plasmas, dense astrophysical plasmas or quark-gluon plasmas. However, experiments have shown that entering the strongly coupled regime is inhibited by disorder- (or correlation-) induced heating which leads to a significant and rapid temperature increase during plasma formation in ultracold systems [3–7]. The process of disorder-induced heating [1,8] can be understood if we consider that ultracold plasmas are commonly created in a state far from equilibrium: During plasma formation the interaction potential switches from a weak Lennard-Jones-like interaction potential between neutral atoms to a repulsive Coulomb potential between the charged ions. Disorder-

induced heating occurs when the Coulomb interaction energy is converted to kinetic energy of the ions during equilibration of the ionic system. The ionic sub-system typically equilibrates within a period of time given by the inverse ion plasma frequency $\omega_i^{-1} = \sqrt{m_i/(4\pi e^2 n_i)}$. Due to ultrafast electron relaxation we can treat the electrons adiabatically during this stage and therefore assume energy conservation for the ionic sub-system,

$$E_i^{\text{total}} = E_i^{\text{pot}} + E_i^{\text{kin}} = \text{const.} \tag{2}$$

Since the ionisation process does not significantly change the kinetic energy of the ions, the equilibration process within the ionic system is driven by changes of the interaction potential. This process finally results in a new equilibrium between the potential and kinetic energies.

The amount of disorder-induced heating can then be estimated by comparing the equilibrium state before and after equilibration of the ionic system: The potential energy can be written in terms of the two-particle correlation function $\xi(r, t)$ as

$$E_i^{\text{corr}} = \frac{n_i}{2} \int d\mathbf{r}\, v(r) \xi(r, t)\,, \tag{3}$$

with the two-particle interaction potential $v(r)$. This correlation energy is then converted to a kinetic energy which is given by $E = \frac{3}{2} k_B T$ for a system in equilibrium. If thermal equilibrium is assumed at initial time $t_i$ of plasma formation and that the plasma reaches equilibrium at a final time $t_f$, the change of the ion temperature $T_i$ during the equilibration process is given by

$$\Delta T_i = T_i(t_f) - T_i(t_i) = \frac{2}{3 k_B} \Delta E_i^{\text{corr}} = \frac{n_i}{3 k_B} \int d\mathbf{r}\, v(r) \big[ \xi(r, t_f) - \xi(r, t_i) \big]\,. \tag{4}$$

It can be seen from (4) that the amount of heating generated depends directly on the difference between the degree of correlation or amount of structure before and after ionisation. Before ionisation the neutral atoms are typically randomly distributed and the correlation energy is negligible. After ionisation the repulsive Coulomb interactions between the ions will impose a structure on the ionic system and therefore the correlation function $\xi(r, t_f)$ will be large compared to the negligible correlations within the initially unordered system. One way to minimise disorder-induced heating is by introducing order into the system prior to ionisation. This can be achieved, for instance, by producing the plasma from a Rydberg blockaded neutral gas with repulsive interactions [1, 9, 10]. The Rydberg blockade [11] introduces a spatial anti-correlation, where the probability to find close-by pairs is strongly suppressed (see Figure 1). The repulsive interaction between the Rydberg atoms increases the resemblance to the structure induced by the Coulomb potential.

In order to determine the amount of disorder-induced heating produced during equilibration, knowledge of the correlation function $\xi(r, t)$ at time $t$ is required. There are numerous methods to obtain the correlation function ranging from numerical many-body molecular dynamics (MD) simulations to analytical methods (see [1] for an extensive review).

Typically, analytical approaches based on the Vlasov or hydrodynamic equations are proposed in order to determine the correlation function. While these approaches describe the system already quite well, they are, strictly speaking, only valid in the continuum (or thermodynamic) limit, i. e. when the number of particles $N \to \infty$. However, if the thermodynamic limit cannot be assumed, as is the case for a dilute gas, the discrete nature of the many-particle system can no longer be neglected. In this work we propose a particle based approach dubbed Kinetic

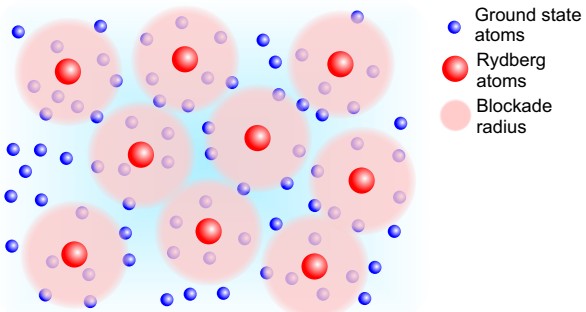

Figure 1: Rydberg blockade in a gas: The ground state atoms (blue) are excited into Rydberg states (red). Due to the Rydberg blockade, there is a blockade radius $R_b$ around each Rydberg atom where no other ground state atom can be excited, giving rise to spatial anti-correlations.

Field Theory (KFT) [12–14], which can be applied to any system of particles obeying the classical Hamiltonian equations of motion and does not rely on the thermodynamic limit. This approach has successfully been used in the context of cosmic large-scale structure formation to obtain the matter-density correlation function in the Universe [12, 13, 15, 16]. Here, we adapt the formalism to describe the evolution of a many-body system of ions produced from a Rydberg blockaded neutral gas. Our results in Sec. 4.5 show that effects due the discrete nature of the particle system, in fact, become visible even at very high packing fractions and have to be taken into account to obtain correct correlation functions which agree with numerical many-body simulations. In addition, no assumption about the system being in thermal equilibrium is required. This allows us to track the disorder-induced heating temperature while the system is still out-of-equilibrium.

We first give a summary of the KFT formalism in Sec. 2. In Sec. 3 we apply the formalism to the ultra-cold plasma system which we want to study. We then provide the results for the obtained correlation functions and disorder-induced temperatures for different sets of parameters in Sec. 4. Finally, we give an outlook on possible extensions of the theoretical framework and offer our conclusions in Sec. 5. Technical details and explicit calculations are provided in the appendices.

## 1.1 Notation

In this work, we will consider the microscopic phase-space coordinates of a large set of $N$ particles confined to a volume $V$. Individual particles are enumerated with *particle labels* $j = 1, \ldots, N$ and have positions and momenta denoted as $d$-dimensional vectors $\vec{q}_j$ and $\vec{p}_j$. We combine the vectors $\vec{q}_j$ and $\vec{p}_j$ into a $2d$-dimensional phase-space coordinate vector

$$\vec{x}_j = \begin{pmatrix} \vec{q}_j \\ \vec{p}_j \end{pmatrix} \quad \forall j \in \{1, \ldots, N\}. \tag{5}$$

These phase-space coordinate vectors are then bundled for all $N$ particles into

$$\boldsymbol{q} = \vec{q}_j \otimes \vec{e}_j, \qquad \boldsymbol{p} = \vec{p}_j \otimes \vec{e}_j, \qquad \boldsymbol{x} = \vec{x}_j \otimes \vec{e}_j, \tag{6}$$

where $\vec{e}_j$ is the canonical base vector in $N$ dimensions with entries $(\vec{e}_j)_i = \delta_{ij}$. The Einstein summation convention is always implied unless explicitly stated otherwise or obvious from context.

The bold vectors follow the rules of matrix multiplication, inducing a scalar product

$$
\boldsymbol{a} \cdot \boldsymbol{b} = \boldsymbol{a}^\top \boldsymbol{b} = \sum_{j=1}^{N} \vec{a}_j^\top \vec{b}_j = \vec{a}_j \cdot \vec{b}_j \, . \tag{7}
$$

If $\boldsymbol{a}, \boldsymbol{b}$ are functions of time, the scalar product includes an additional integration over the time argument,

$$
\boldsymbol{a} \cdot \boldsymbol{b} = \int_{t_\mathrm{i}}^{t_\mathrm{f}} \mathrm{d}t \, \boldsymbol{a}(t) \cdot \boldsymbol{b}(t) \, . \tag{8}
$$

## 2 The analytical framework of Kinetic Field Theory

### 2.1 Generating functional

KFT [12–14] is based on the path-integral formulation for classical mechanics [17]. It contains both the dynamics and the initial statistics of a classical non-equilibrium many-particle system in a generating functional. Time evolution is encoded by a functional integral over all possible phase-space trajectories of the individual particles. Structurally, KFT resembles a non-equilibrium quantum field theory but simplifies considerably due to the symplectic structure of Hamilton's equations of motion and the deterministic nature of classical particle-dynamics. Since we are dealing with classical particles, a functional delta distribution is introduced into the generating functional that assigns a non-vanishing weight only to those trajectories which obey the classical equations of motion. The stochastic element is introduced by integrating over an initial phase-space probability distribution $\mathcal{P}(\boldsymbol{x}^{(\mathrm{i})})$. The generating functional is thus given by

$$
Z = \int \mathrm{d}\boldsymbol{x}^{(\mathrm{i})} \, \mathcal{P}(\boldsymbol{x}^{(\mathrm{i})}) \int_{\boldsymbol{x}^{(\mathrm{i})}} \mathcal{D}\boldsymbol{x}(t) \, \delta_\mathrm{D}\big[\boldsymbol{E}[\boldsymbol{x}(t)]\big] \, , \tag{9}
$$

where $\boldsymbol{E}[\boldsymbol{x}(t)] = 0$ is the equation of motion for the $N$-particle system. For later convenience, we express the functional delta distribution in (9) as a Fourier transform, which introduces the auxiliary field $\boldsymbol{\chi}(t)$ as conjugate to the phase-space coordinate $\boldsymbol{x}(t)$ and write the generating functional as (9)

$$
Z = \int \mathrm{d}\boldsymbol{x}^{(\mathrm{i})} \, \mathcal{P}(\boldsymbol{x}^{(\mathrm{i})}) \int_{\boldsymbol{x}^{(\mathrm{i})}} \mathcal{D}\boldsymbol{x}(t) \int \mathcal{D}\boldsymbol{\chi}(t) \, \mathrm{e}^{\mathrm{i}\boldsymbol{\chi} \cdot \boldsymbol{E}[\boldsymbol{x}(t)]} \, . \tag{10}
$$

We describe phase-space dynamics by Hamilton's equations of the form

$$
0 = \boldsymbol{E}(\boldsymbol{x}) = \dot{\boldsymbol{x}} - \mathscr{I} \nabla_{\boldsymbol{x}} \mathcal{H}(\boldsymbol{x}) \, , \quad \mathscr{I} := \begin{pmatrix} 0 & \mathcal{I}_3 \\ -\mathcal{I}_3 & 0 \end{pmatrix} \otimes \mathcal{I}_N \, , \tag{11}
$$

where $\mathscr{I}$ is the usual symplectic matrix and $\mathcal{I}_d$ denotes the $d$-dimensional identity matrix. The Hamiltonian can be split into the free part $\mathcal{H}_0$ which describes the free (or inertial) motion of particles and an interacting part $\mathcal{H}_\mathrm{I}$. We restrict ourselves to systems of $N$ identical particles in the absence of external forces and assume that the force between particles acts instantaneously and

only depends on their configuration-space positions. We can then express $\mathcal{H}_{\mathrm{I}}$ as a superposition of $N$ single particle potentials $v$,

$$\mathcal{H}_{\mathrm{I}}(\boldsymbol{q}(t)) := \frac{1}{2} \sum_{j \neq k=1}^{N} v(|\vec{q}_j - \vec{q}_k|, t) \,. \tag{12}$$

Just as for the Hamiltonian, we can split the action into a free and an interacting part, $S = S_0 + S_{\mathrm{I}}$, where

$$S_0 := \boldsymbol{\chi} \cdot (\dot{\boldsymbol{x}} - \mathcal{I} \nabla_{\boldsymbol{x}} \mathcal{H}_0(\boldsymbol{x})) \quad \text{and} \quad S_{\mathrm{I}} := \boldsymbol{\chi}_{\boldsymbol{p}} \cdot \nabla_q \mathcal{H}_{\mathrm{I}} \,. \tag{13}$$

Following [12], we can then write the generating functional as

$$Z = \int \mathrm{d}\Gamma_{\mathrm{i}} \int_{\boldsymbol{x}^{(\mathrm{i})}} \mathcal{D}\boldsymbol{x}(t) \int \mathcal{D}\boldsymbol{\chi}(t) \, \mathrm{e}^{\mathrm{i}S_0 + \frac{\mathrm{i}}{2} \Phi \cdot \sigma \cdot \Phi} \,. \tag{14}$$

For a compact notation we have defined the interaction matrix

$$\sigma(\boldsymbol{x}, t; \boldsymbol{x}', t') = -\delta_{\mathrm{D}}(t - t') \, v(|\vec{q} - \vec{q}'|) \begin{pmatrix} 0 & 1 \\ 1 & 0 \end{pmatrix}, \tag{15}$$

containing the single-particle potential $v$ and we abbreviate the integral over the initial phase-space distribution as $\mathrm{d}\Gamma_{\mathrm{i}} = \mathrm{d}\boldsymbol{x}^{(\mathrm{i})} \mathcal{P}(\boldsymbol{x}^{(\mathrm{i})})$. We have furthermore introduced the tuple

$$\Phi = \begin{pmatrix} \Phi_f \\ \Phi_B \end{pmatrix} \tag{16}$$

of two collective fields: the Klimontovich phase-space density $\Phi_f$ and the response field $\Phi_B$ given by

$$\Phi_f(\vec{x}, t) = \sum_{j=1}^{N} \delta_{\mathrm{D}}(\vec{x} - \vec{x}_j(t)) \quad \text{and} \quad \Phi_B(\vec{x}, t) = \sum_{j=1}^{N} \vec{\chi}_{p_j}(t) \cdot \nabla_q \delta_{\mathrm{D}}(\vec{x} - \vec{x}_j(t)). \tag{17}$$

The phase-space density $\Phi_f$ contains the information on which particles of the ensemble occupy a phase-space state $\vec{x}$ at time $t$. The response field $\Phi_B$ encodes the deviation of all individual particles from their inertial trajectories as a linear response to a disturbance, which is caused by the interaction with all other particles. With the vector tuple $\vec{s} = (\vec{k}, \vec{l})$ denoting the Fourier conjugate to $\vec{x} = (\vec{q}, \vec{p})$, the Klimontovich phase-space density and response field in Fourier space read

$$\Phi_f(\vec{s}, t) = \int \mathrm{d}^6 x \, \mathrm{e}^{-\mathrm{i}\vec{s} \cdot \vec{x}} \sum_{j=1}^{N} \delta_{\mathrm{D}}(\vec{x} - \vec{x}_j(t)), \tag{18}$$

$$\Phi_B(\vec{s}, t) = \int \mathrm{d}^6 x \, \mathrm{e}^{-\mathrm{i}\vec{s} \cdot \vec{x}} \sum_{j=1}^{N} \vec{\chi}_{p_j}(t) \cdot \nabla_q \delta_{\mathrm{D}}(\vec{x} - \vec{x}_j(t)). \tag{19}$$

Next we introduce source fields $H_f$ and $H_B$ which couple to the collective fields $\Phi_f$ and $\Phi_B$, as well as source fields $J$ and $K$ coupling to $\boldsymbol{x}$ and $\boldsymbol{\chi}$, respectively. By replacing

$$\vec{x}_j(t) \to \hat{\vec{x}}_j(t) := \frac{\delta}{\mathrm{i}\delta \vec{J}_j(t)} \quad \text{and} \quad \vec{\chi}_j(t) \to \hat{\vec{\chi}}_j(t) := \frac{\delta}{\mathrm{i}\delta \vec{K}_j(t)} \tag{20}$$

in the collective fields $\Phi$ we turn them into operators $\hat{\Phi}$ which, in Fourier space, take the form

$$\hat{\Phi}_f(1) := \sum_{j=1}^{N} \exp\left(-i\vec{s}_1 \cdot \frac{\delta}{i\delta \vec{J}_{x_j}(t_1)}\right) \quad \text{and} \quad \hat{\Phi}_B(1) := \sum_{j=1}^{N} \left(i\vec{k}_1 \cdot \frac{\delta}{i\delta \vec{K}_{p_j}(t_1)}\right)\hat{\Phi}_{f_j}(1)\,, \qquad (21)$$

where we used the notation $(1) := (t_1, \vec{s}_1)$. This allows us to re-write the generating functional as

$$
\begin{aligned}
Z[H, J, K] &= e^{\frac{i}{2}\hat{\Phi}\cdot\sigma\cdot\hat{\Phi}}\, e^{iH\cdot\hat{\Phi}} \int d\Gamma_i \int_{x^{(i)}} \mathcal{D}x \int \mathcal{D}\chi\, e^{iS_0 + J\cdot x + K\cdot\chi} \\
&= e^{\frac{i}{2}\hat{\Phi}\cdot\sigma\cdot\hat{\Phi}}\, e^{iH\cdot\hat{\Phi}} Z_0[J, K] \\
&=: e^{i\hat{S}_I} Z_0[H, J, K]\,.
\end{aligned}
\qquad (22)
$$

where we have introduced the *free* generating functional $Z_0[H, J, K]$ and the interaction operator $\hat{S}_I = \frac{1}{2}\hat{\Phi}\cdot\sigma\cdot\hat{\Phi}$. The form of the generating functional in (22) now allows us to obtain cumulants (i. e. the connected part of correlation functions) of the collective fields by taking appropriate functional derivatives of the logarithm of the generating functional with respect to the source fields,

$$
\begin{aligned}
G_{\alpha_1 \dots \alpha_n}(1, \dots, n) &:= \left\langle \Phi_{\alpha_1}(1) \dots \Phi_{\alpha_n}(n) \right\rangle_{\text{connected}} \\
&= \frac{\delta}{i\delta H_{\alpha_1}(1)} \cdots \frac{\delta}{i\delta H_{\alpha_n}(n)} \ln Z[H]\bigg|_{H=0}\,.
\end{aligned}
\qquad (23)
$$

We have shown in [12] that the path integrals in the *free* generating functional $Z_0[J, K]$ can be performed once the Green's function $\mathcal{G}$ of the free equation of motion of a single particle is known. One finds

$$Z_0[J, K] = \int dx^{(i)}\, \mathcal{P}(x^{(i)})\, e^{iJ\cdot\bar{x}}\,, \qquad (24)$$

where the solution to the equations of motion $\bar{x}(t)$ is defined as

$$\bar{x}(t) = G(t, t_i)x^{(i)} - \int_{t_i}^{t_f} dt'\, G(t, t')K(t')\,, \qquad (25)$$

where $G(t, t_i)x^{(i)}$ is the homogeneous solution and the Green's function has the general form

$$G(t, t') = \mathcal{G}(t, t') \otimes \mathcal{I}_N \quad \text{with} \quad \mathcal{G}(t, t') = \begin{pmatrix} g_{qq}(t, t')\mathcal{I}_d & g_{qp}(t, t')\mathcal{I}_d \\ g_{pq}(t, t')\mathcal{I}_d & g_{pp}(t, t')\mathcal{I}_d \end{pmatrix}\,. \qquad (26)$$

The components of the Green's function and the initial probability distribution $\mathcal{P}(x^{(i)})$ will be specified later in Section 3.1. The additional source term $K$ in (25) allows us to deflect particles from their inertial motion if particles are interacting via an interaction potential. For freely streaming particles we can simply set $K = 0$. For interacting particles, however, the collective-field cumulants in (23) can only be computed perturbatively. A naive perturbative approach is to expand the exponential interaction operator in (22) in orders of the interaction matrix $\sigma$. We thus evaluate the force exerted on a particle along its inertial trajectory due to the interactions with all other

particles on their respective inertial trajectories. Since the auxiliary fields $\vec{\chi}_j$ in the response field (19) are replaced by functional derivatives with respect to $\vec{K}_j$, this force is then inserted for $\vec{K}_j$ in (25) and modifies the inertial trajectory of that particle. The perturbed physical quantities in the perturbation theory in KFT are therefore the trajectories of particles.

It is clear that in this naive approach any low-order truncation of the perturbation series will result in only small corrections to the free evolution of particles. It is for that reason that we introduce a resummed version of KFT in the next chapter which allows us to set up a more appropriate perturbation theory. The Resummed Kinetic Field Theory (RKFT) then allows us to capture a significant part of the effects caused by the interaction potential already at low perturbative orders. For the reader familiar with RKFT, we briefly recap the formalism in Sec. 2.2. The interested reader yet unfamiliar with RKFT we refer to [15] for a comprehensive derivation of the formalism.

## 2.2 Resummed Kinetic Field Theory

To obtain the generating functional of RKFT we reformulate the path integral in terms of the actual macroscopic fields of interest rather than the underlying microscopic fields $x$ and $\chi$. This will allow us to set up a new perturbative approach to KFT in terms of propagators and vertices following the standard procedure known from quantum and statistical field theory.

We briefly summarise the procedure detailed in [15] which leads to the new form of the generating functional: We begin by collecting the microscopic fields $x_j$ and $\chi_j$ into a tuple, $\psi_j := (x_j, \chi_j)$, and reformulate the generating functional (14) in terms of the macroscopic field $f$ by introducing a functional delta distribution to replace the explicitly $\psi$-dependent field $\Phi_f[\psi]$ by the formally $\psi$-independent field $f$,

$$Z = \int d\Gamma_i \int_{x^{(i)}} \mathcal{D}\psi \int \mathcal{D}f \ e^{iS_0[\psi] + if \cdot \sigma_{fB} \cdot \Phi_B[\psi]} \delta_D[f - \Phi_f[\psi]] . \tag{27}$$

In a second step, we introduce a conjugate field $\beta$ to express the delta distribution by its functional Fourier transform in (27),

$$Z = \int \mathcal{D}f \int \mathcal{D}\beta \ e^{-i\beta \cdot f} \int d\Gamma_i \int_{x^{(i)}} \mathcal{D}\psi \ e^{iS_0[\psi] + i\beta \cdot \Phi_f[\psi] + if \cdot \sigma_{fB} \cdot \Phi_B[\psi]} \tag{28}$$

$$= \int \mathcal{D}\phi \ e^{-i\beta \cdot f} Z_{\Phi,0}[\tilde{\phi}] . \tag{29}$$

The auxiliary field $\beta$ is conjugate to $f$ in the same way as $\chi_j$ is to $x_j$. As before, $S_0[\psi]$ denotes the action of the free theory. We defined the macroscopic field tuples $\phi := (f, \beta)$ and $\tilde{\phi} := (\beta, f \cdot \sigma_{fB})$ for a compact notation. Finally, using $\ln Z_{\Phi,0}[\tilde{\phi}] = W_{\Phi,0}[\tilde{\phi}]$, we arrive at the desired form for the generating functional,

$$Z = \int \mathcal{D}\phi \ e^{iS_\phi[\phi]} , \tag{30}$$

where the full macroscopic action is defined as

$$iS_\phi[\phi] := -if \cdot \beta + W_{\Phi,0}[\tilde{\phi}] . \tag{31}$$

The microscopic information of the system is now encoded in the free generating functional $W_{\Phi,0}$ which can be expressed in terms of the *dressed* free cumulants,

$$G^{(0)}_{f\cdots f\mathcal{F}\cdots\mathcal{F}}(1,\ldots,n_f,1',\ldots,n'_{\mathcal{F}}) := \prod_{r=1}^{n_{\mathcal{F}}}\left(\int d\bar{r}\;\sigma_{fB}(r',-\bar{r})\right)$$
$$\times\, G^{(0)}_{f\cdots fB\cdots B}(1,\ldots,n_f,\bar{1},\ldots,\bar{n}_{\mathcal{F}})\,, \tag{32}$$

by means of a functional Taylor expansion. It then reads

$$W_{\Phi,0}[\tilde{\phi}] := \sum_{n_\beta,n_f=0}^{\infty}\frac{i^{n_\beta+n_f}}{n_\beta!\,n_f!}\prod_{u=1}^{n_\beta}\left(\int du\,\beta(-u)\right)\prod_{r=1}^{n_f}\left(\int dr'\,f(-r')\right)$$
$$\times\, G^{(0)}_{f\cdots f\mathcal{F}\cdots\mathcal{F}}(1,\ldots,n_\beta,1',\ldots,n'_f)\,. \tag{33}$$

The reformulation of KFT in (30) is exact and the macroscopic action (31) can be understood as an exact *effective action* of the theory. It is an effective action in the sense that in (33) we integrate over all microscopic degrees of freedom. It is exact because the evolution of the macroscopic fields is governed by the underlying microscopic dynamics.

The new path integral in (30) allows us to set up a new perturbation scheme in terms of vertices and propagators following the standard procedure of quantum and statistical field theory. As shown in [15] we can split the action (31) into a propagator part $S_\Delta[\phi]$ which contains all terms quadratic in $\phi$ and a vertex part $S_{\mathcal{V}}[\phi]$ that contains all of the remaining terms $S_{\mathcal{V}}[\phi]$,

$$iS_\phi[\phi] \stackrel{!}{=} iS_\Delta[\phi] + iS_{\mathcal{V}}[\phi] \tag{34}$$

$$:= -\frac{1}{2}\int d1\int d2\,(f,\beta)(-1)\begin{pmatrix}(\Delta^{-1})_{ff} & (\Delta^{-1})_{f\beta}\\(\Delta^{-1})_{\beta f} & (\Delta^{-1})_{\beta\beta}\end{pmatrix}(1,2)\begin{pmatrix}f\\\beta\end{pmatrix}(-2)$$

$$+\sum_{\substack{n_\beta,n_f=0\\n_\beta+n_f\neq 2}}^{\infty}\frac{1}{n_\beta!\,n_f!}\prod_{u=1}^{n_\beta}\left(\int du\,\beta(-u)\right)\prod_{r=1}^{n_f}\left(\int dr'\,f(-r')\right) \tag{35}$$

$$\times\,\mathcal{V}_{\beta\cdots\beta f\cdots f}(1,\ldots,n_\beta,1',\ldots,n'_f)\,,$$

where we introduced the inverse macroscopic propagator $\Delta^{-1}$ and the macroscopic $(n_\beta+n_f)$-point vertices $\mathcal{V}_{\beta\cdots\beta f\cdots f}$.

The macroscopic generating functional $Z_\phi$ can now be defined by introducing a source field $M=(M_f,M_\beta)$ conjugate to $\phi$ into the partition function,

$$Z_\phi[M] := \int\mathcal{D}\phi\;e^{iS_\phi[\phi]+iM\cdot\phi}\,. \tag{36}$$

The vertex part of the action can be pulled in front of the path integral by replacing its $\phi$-dependence with functional derivatives with respect to $M$, $\hat{S}_{\mathcal{V}} := S_{\mathcal{V}}\left[\frac{\delta}{i\delta M}\right]$, acting on the remaining path integral,

$$Z_\phi[M] = e^{i\hat{S}_{\mathcal{V}}}\int\mathcal{D}\phi\;e^{-\frac{1}{2}\phi\cdot\Delta^{-1}\cdot\phi+iM\cdot\phi} \tag{37}$$

$$= e^{i\hat{S}_{\mathcal{V}}}\,e^{\frac{1}{2}(iM)\cdot\Delta\cdot(iM)}\,. \tag{38}$$

Going from (37) to (38) the functional determinant was absorbed into the normalisation of the path integral.

Expanding the first exponential in orders of the vertices then gives rise to the new macroscopic perturbation theory. Within this approach, the interacting two-point phase-space density cumulant $G_{ff}$, for example, is obtained via

$$
\begin{aligned}
G_{ff}(1,2) &= \frac{\delta}{\mathrm{i}\delta M_f(1)} \frac{\delta}{\mathrm{i}\delta M_f(2)} \ln Z_\phi[M]\bigg|_{M=0} \\
&= \Delta_{ff}(1,2) + \text{terms involving vertices}.
\end{aligned}
\tag{39}
$$

We will refer to $\Delta$ as the tree-level result of the two-point phase-space density cumulant $G_{ff}$. The terms containing vertices will thus provide self-energy (or loop) corrections to the tree-level result. The inverse propagators $\Delta^{-1}$ and the vertices $\mathcal{V}_{\beta\cdots\beta f\cdots f}$ in (39) can be identified as

$$
\Delta^{-1}(1,2) = \begin{pmatrix} (\Delta^{-1})_{ff} & (\Delta^{-1})_{f\beta} \\ (\Delta^{-1})_{\beta f} & (\Delta^{-1})_{\beta\beta} \end{pmatrix}(1,2) = \begin{pmatrix} \sigma_{fB} \cdot G_{BB}^{(0)} \cdot \sigma_{Bf} & \mathrm{i}\mathcal{I} + \sigma_{fB} \cdot G_{Bf}^{(0)} \\ \mathrm{i}\mathcal{I} + G_{fB}^{(0)} \cdot \sigma_{Bf} & G_{ff}^{(0)} \end{pmatrix}(1,2),
\tag{40}
$$

$$
\mathcal{V}_{\beta\cdots\beta f\cdots f}(1,\ldots,n_\beta,1',\ldots,n_f') = \mathrm{i}^{n_\beta+n_f} \prod_{r=1}^{n_f}\left(\int \mathrm{d}\bar{r}\; \sigma_{fB}(r',-\bar{r})\right) G_{f\cdots fB\cdots B}^{(0)}(1,\ldots,n_\beta,\bar{1},\ldots,\bar{n}_f),
\tag{41}
$$

where $\mathcal{I}$ denotes the identity two-point function,

$$
\mathcal{I}(1,2) := (2\pi)^3 \delta_{\mathrm{D}}\big(\vec{k}_1 + \vec{k}_2\big)\delta_{\mathrm{D}}(t_1 - t_2).
\tag{42}
$$

The propagator $\Delta$ is obtained by a combined matrix and functional inversion of $\Delta^{-1}$, defined via the matrix integral equation

$$
\int \mathrm{d}\bar{1}\; \Delta(1,-\bar{1})\, \Delta^{-1}(\bar{1},2) \overset{!}{=} \mathcal{I}(1,2)\, \mathcal{I}_2,
\tag{43}
$$

with the $2\times 2$ unit matrix $\mathcal{I}_2$. While the matrix inversion can be performed immediately and yields

$$
\Delta(1,2) = \begin{pmatrix} \Delta_{ff} & \Delta_{f\beta} \\ \Delta_{\beta f} & \Delta_{\beta\beta} \end{pmatrix}(1,2) = \begin{pmatrix} \Delta_{\mathrm{R}} \cdot G_{ff}^{(0)} \cdot \Delta_{\mathrm{A}} & -\mathrm{i}\Delta_{\mathrm{R}} \\ -\mathrm{i}\Delta_{\mathrm{A}} & 0 \end{pmatrix}(1,2),
\tag{44}
$$

the remaining functional inversions,

$$
\Delta_{\mathrm{R}}(1,2) = \Delta_{\mathrm{A}}(2,1) := \left(\mathcal{I} - \mathrm{i}G_{fB}^{(0)} \cdot \sigma_{Bf}\right)^{-1}(1,2),
\tag{45}
$$

have to be computed numerically in general. The numerical evaluation, however, is inexpensive and is explained in great detail in [15]. The computation of the self-energy (or loop) contributions in (39) up to one-loop order is detailed in Appendix B.

## 3 Treatment of an ultracold ion plasma from an initially correlated Rydberg gas in the framework of RKFT

Our first goal is to compute the two-point correlation function $\xi(r)$ for a system of ultracold, interacting and initially correlated ions using RKFT and to directly compare our analytical results

to numerical molecular dynamics simulations. Our second goal is to use the correlation functions provided by RKFT to predict the temperature rise due to disorder-induced heating within the ion plasma. In the following we set up the properties of our system of ultracold particles and their description in the formalism of (R)KFT. Our general assumptions for the ultracold system are

(1) We assume that prior to ionisation our system consists of atoms in a Rydberg state. We do not make any assumptions about the specific excitation mechanism and treat the Rydberg atoms as hard spheres. At this point we determine our initial conditions (see Section 3.2).

(2) We assume that at some initial time $t_i$ the Rydberg atoms are all ionised instantaneously. We do not make any assumptions about the ionisation mechanism. This marks the starting point for the evolution of our ionic system. At this point the ions start to interact with each other via a simple potential described further in Section 3.1.

(3) We restrict our calculations to only one particle species and thus only consider the evolution of the ions. Although we neglect the presence of the free electrons as a separate particle species in the plasma, our interaction potential has an exponential cut-off which approximates Debye shielding.

These simplifying assumptions allow us to demonstrate and constrain the predictive power of RKFT and also discuss limitations of the theoretical framework developed so far. We present our results in Section 4 and discuss in Section 5 how the above assumptions can be relaxed in order to describe more realistic settings.

## 3.1 Equations of motion and interaction potential for the ultracold plasma

We will treat the ions in the plasma as well as the Rydberg atoms at initial time as classical particles. This is possible for the Rydberg atoms since quantum mechanical properties of the Rydberg excitation only play a role for generating the initial state of our system. For our description within RKFT it does, however, not matter how the initial conditions were established as long as they fulfil a few conditions further discussed in Section 3.2. The separations between Rydberg atoms at initial times are sufficiently large so that quantum effects can safely be neglected and we can treat the Rydberg gas as a purely classical system of $N$ particles of mass $m$. For the ions we also assume that their separations remain sufficiently large during the evolution of the system so that any quantum mechanical effects can be neglected.

We can then safely assume that the dynamics of the system is governed by the simple Hamiltonian equations of motion given by

$$\dot{\vec{q}}_j = \frac{\vec{p}_j}{m},\tag{46}$$

$$\dot{\vec{p}}_j = -\sum_{\substack{k=1 \\ k \neq j}}^{N} \vec{\nabla}_{q_j} v\big(|\vec{q}_j(t) - \vec{q}_k(t)|\big),\tag{47}$$

which in the non-interacting case, $v = 0$, are solved according to (25) by

$$\begin{pmatrix} \vec{q}_j(t) \\ \vec{p}_j(t) \end{pmatrix} = \begin{pmatrix} g_{qq}(t,t')\mathcal{I}_3 & g_{qp}(t,t')\mathcal{I}_3 \\ g_{pq}(t,t')\mathcal{I}_3 & g_{pp}(t,t')\mathcal{I}_3 \end{pmatrix} \begin{pmatrix} \vec{q}_j^{(i)} \\ \vec{p}_j^{(i)} \end{pmatrix}\tag{48}$$

with the components of the matrix valued Green's function (26) given by

$$g_{qq}(t,t') = g_{pp}(t,t') = \Theta(t-t'), \tag{49}$$

$$g_{qp}(t,t') = \frac{t-t'}{m} \Theta(t-t'), \tag{50}$$

$$g_{pq}(t,t') = 0. \tag{51}$$

After ionisation the ionized Rydberg atoms still follow trajectories described by the equations of motion (46) and (47). The interaction potential in the ion plasma would transition to a Coulomb potential with Debye shielding which results in a Yukawa potential. However, both the Lennard-Jones and the Yukawa potential have a non-integrable singularity at $r = 0$. Since we will compare our analytical results to simulation results later on, a softening scale would have to be introduced in the denominator in order to evaluate the forces acting on particles [1]. This would introduce a rather arbitrary free parameter into the theory which we would like to avoid at this point. For simplicity, we therefore model the ion-ion interactions by a repulsive potential of Gaussian form,

$$\tilde{v}_{\mathrm{G}}(k) = A(2\pi\sigma^2)^{\frac{3}{2}} e^{-\frac{\sigma^2 k^2}{2}}, \tag{52}$$

and furthermore assume that the potential at initial time $t_{\mathrm{i}}$, when the system is still in a Rydberg state, is the same as the potential at final time $t_{\mathrm{f}}$ in the ionic system. Similarly to a Yukawa potential, we can easily control the range and gradient of the interaction potential by changing the width of the Gaussian $\sigma$ without having to change its form. The choice of a repulsive potential for the initial Rydberg system is motivated by the finding that a repulsive Coulomb-like interaction potential helps to increase the order in the initial Rydberg system, which further decreases disorder-induced heating in an ultracold plasma. Such a repulsive potential can be realised in experiments [2].

## 3.2 Initial conditions from the ultracold Rydberg system

The formation of the blockade radius around Rydberg atoms induces a natural anti-correlation in a Rydberg gas [18–22]. We use the resulting anti-correlation function as the initial conditions for our system. We furthermore assume that we can neglect any initial momentum-momentum and position-momentum correlations. Thus, we include only position-position correlations and a momentum dispersion (i.e. momentum auto-correlations) which is due to a physical temperature that is set externally. We also assume that – averaged over sufficiently large scales – the system is homogeneous and isotropic.

To set up the initial correlation function we consider the following scenario: $N$ ground-state atoms with a high packing fraction are simultaneously excited into the Rydberg state. Due to the Rydberg blockade no two Rydberg atoms can be closer than $2R_b$. Within this picture we can thus well approximate the Rydberg atoms as hard spheres.

We assume that any realisation (or configuration) of $N$ Rydberg atoms is equally likely. The initial density field of Rydberg atoms can thus be described as a statistically homogeneous and isotropic Gaussian random field. It is completely characterised by the mean density and the two-point correlations induced by the Rydberg blockade. We can then obtain the initial conditions for

---

[1] In addition, we require the softening scale to perform the Fourier transforms of the interaction potentials as input for the RKFT calculations. Technically, this is only necessary for the Lennard-Jones potential. For the Yukawa potential the smoothing scale can be set to zero after the transformation.

the positions for the Rydberg particles by a Poisson sampling-process of that initial density field. The initial momenta are independently sampled from a Gaussian momentum distribution. The initial phase-space probability distribution function for this case have been derived in [12] and is given by

$$\mathcal{P}(\boldsymbol{x}^{(\mathrm{i})}) = \frac{V^{-N}}{\sqrt{(2\pi)^{3N}\det C_{pp}}} \mathcal{C}(\boldsymbol{p}) \exp\left(-\frac{1}{2}\boldsymbol{p}^{\top}C_{pp}^{-1}\boldsymbol{p}\right), \tag{53}$$

where $\boldsymbol{p} = \vec{p}_j \otimes \vec{e}_j$ and where we have dropped the superscript (i) for $q$ and $p$ for better readability. The correlation operator $\mathcal{C}(\boldsymbol{p})$ appearing in (53) contains the correlation matrices for density-density and momentum-momentum correlations.

For a full derivation of the expression we refer the reader to [12]. Here, we only provide the final expression for the correlation operator given by

$$\mathcal{C}(\boldsymbol{p}) = (-1)^N \prod_{j=1}^{N}\left(1 + \left(C_{\delta p}\frac{\partial}{\partial \boldsymbol{p}}\right)_j\right) + (-1)^N \sum_{(j,k)}(C_{\delta\delta})_{jk}\prod_{\{l\}'}\left(1 + \left(C_{\delta p}\frac{\partial}{\partial \boldsymbol{p}}\right)_l\right)$$

$$+ (-1)^N \sum_{(j,k)}(C_{\delta\delta})_{jk}\sum_{(a,b)'}(C_{\delta\delta})_{ab}\prod_{\{l\}''}\left(1 + \left(C_{\delta p}\frac{\partial}{\partial \boldsymbol{p}}\right)_l\right) + \dots \tag{54}$$

where $j \neq k$ as well as $a \neq b$ and $\{\}'$ indicates that $l$ runs over all indices except $(j,k)$ and $\{\}''$ indicates that $l$ runs over all indices except $(j,k,a,b)$. The correlation matrices in (54) are defined as follows,

$$C_{\delta\delta} := \langle \delta_j \delta_k \rangle = \int \frac{\mathrm{d}^3 k}{(2\pi)^3} P_\delta^{(\mathrm{i})}(k)\mathrm{e}^{-\mathrm{i}\vec{k}\cdot(\vec{q}_j - \vec{q}_k)}, \quad C_{\delta p} := \langle \delta_j \vec{p}_k \rangle \quad \text{and}$$

$$C_{pp} := \langle \vec{p}_j \otimes \vec{p}_k \rangle, \tag{55}$$

where the position-position correlation $C_{\delta\delta}$ is defined in terms of its Fourier transform, the power spectrum $P_\delta^{(\mathrm{i})}(k)$, for later convenience. Since we assume that initial position-momentum and momentum-momentum correlations can be neglected, we can set $C_{\delta p} = 0$ and are left with only the diagonal elements of the momentum-correlation matrix $C_{pp}$ which describe the initial velocity dispersion of the particles. The momentum-correlation matrix thus reduces to

$$C_{pp} = \frac{\sigma_\mathrm{p}^2}{3}\mathcal{I}_3 \otimes \mathcal{I}_N, \tag{56}$$

with the thermal momentum dispersion $\sigma_\mathrm{p} = \sqrt{3mk_\mathrm{B}T}$.

## 3.3 Computation of two-point free cumulants

The initial conditions given in (53) enter our analytical RKFT computations via the free cumulants $G^{(0)}_{f\dots f\mathcal{F}\dots\mathcal{F}}(1,\dots,n_f,\bar{1},\dots,\bar{n}_{\mathcal{F}})$. In [14, 15] we have shown how the free cumulants of the theory can be evaluated exactly and use the results here. Since we neglect initial momentum and momentum-density correlations between different particles and only keep momentum auto-correlations for our system of Rydberg atoms, the expression for the initial conditions and subsequently the computation of the free cumulants simplify considerably. As we are only interested in spatial density correlations, the phase-space cumulants reduce to spatial density and response field

cumulants $G^{(0)}_{\rho\cdots\rho\mathcal{F}\cdots\mathcal{F}}(1,\ldots,n_\rho,\bar{1},\ldots,\bar{n}_\mathcal{F})$. The expressions to be evaluated for the non-vanishing dressed free 1- and two-point cumulants, for instance, read

$$G^{(0)}_\rho(1) = (2\pi)^3\delta_{\mathrm{D}}\big(\vec{k}_1\big)\bar{\rho}\,, \tag{57}$$

$$G^{(0)}_{\rho\mathcal{F}}(1,2) = (2\pi)^3\delta_{\mathrm{D}}\big(\vec{k}_1+\vec{k}_2\big)\bar{\rho}\,\mathrm{i}k_1^2\,\nu(k_1)\left(\frac{t_1-t_2}{m}\right)$$
$$\times\exp\left(-\frac{mk_{\mathrm{B}}T_{\mathrm{i}}}{2}\left(\vec{k}_1\frac{t_1-t_2}{m}\right)^2\right)\Theta(t_1-t_2)\,, \tag{58}$$

$$G^{(0)}_{\rho\rho}(1,2) = (2\pi)^3\delta_{\mathrm{D}}\big(\vec{k}_1+\vec{k}_2\big)\left[\bar{\rho}\,\exp\left(-\frac{mk_{\mathrm{B}}T_{\mathrm{i}}}{2}\left(\vec{k}_1\frac{t_1-t_2}{m}\right)^2\right)\right.$$
$$\left.+\bar{\rho}^2\,C_2(1,2)\,\exp\left(-\frac{mk_{\mathrm{B}}T_{\mathrm{i}}}{2}\left(\left(\vec{k}_1\frac{t_1}{m}\right)^2+\left(-\vec{k}_1\frac{t_2}{m}\right)^2\right)\right)\right]\,. \tag{59}$$

The function $C_2(1,2)$ appearing in (59) contains the correlations between two freely evolving particles derived in [14] and is given by

$$C_2(1,2) = \int\mathrm{d}^3q_{12}\,\mathrm{e}^{-\mathrm{i}\vec{k}_1\cdot q_{12}}\left[\left(1+C_{\delta_1\delta_2}-\mathrm{i}C_{\delta_1p_2}\cdot\vec{k}_1\left(\frac{t_1+t_2}{m}\right)\right.\right.$$
$$\left.\left.-\left(C_{\delta_1p_2}\cdot\frac{\vec{k}_1}{m}\right)^2 t_1 t_2\right)\mathrm{e}^{\frac{t_1 t_2}{m^2}\vec{k}_1^\top C_{p_1p_2}\vec{k}_1}-1\right]\,. \tag{60}$$

For the initial state of our system we consider density-density correlations only as discussed in 3.2. In the absence of momentum-momentum and momentum-density correlations (60) reduces to

$$C_2(1,2) = P^{(\mathrm{i})}_\delta(k_1)\,. \tag{61}$$

The expression for the two-point density cumulant (59) then simplifies to

$$G^{(0)}_{\rho\rho}(1,2) = (2\pi)^3\delta_{\mathrm{D}}\big(\vec{k}_1+\vec{k}_2\big)\left[\bar{\rho}\,\exp\left(-\frac{mk_{\mathrm{B}}T_{\mathrm{i}}}{2}\left(\vec{k}_1\frac{t_1-t_2}{m}\right)^2\right)\right.$$
$$\left.+\bar{\rho}^2\,P^{(\mathrm{i})}_\delta(k_1)\,\exp\left(-\frac{mk_{\mathrm{B}}T_{\mathrm{i}}}{2}\left(\left(\vec{k}_1\frac{t_1}{m}\right)^2+\left(-\vec{k}_1\frac{t_2}{m}\right)^2\right)\right)\right]\,. \tag{62}$$

The initial power spectrum $P^{(\mathrm{i})}_\delta(k_1)$ appearing in the free cumulants of RKFT is obtained by a Fourier transform of the initial (anti-)correlation function of the Rydberg gas which is induced by the Rydberg blockade.

The full two-point density cumulant $G_{\rho\rho}$ is then computed according to (39) and has to be evaluated numerically. In Appendix A we provide a simplified example which is closely related to our many-body system of Rydberg atoms and for which the tree-level result of $G_{\rho\rho}$ can be computed analytically. In configuration space, the two-point density cumulant $G_{\rho\rho}$ directly corresponds to the two-point correlation function $\xi(r)$ which is used for the computation of disorder-induced heating.

It is worthwhile to note here that the first term of the free two-point cumulant $G^{(0)}_{\rho\rho}(1,2)$ in (62) which scales linearly with the mean density $\bar{\rho}$ is in fact a shot noise contribution which – as we shall see later – cannot be ignored.

The expressions for the free cumulants for the Rydberg system additionally needed for one-loop contributions are provided in Appendix B.

# 4 The evolution of structure and disorder-induced heating in an ultracold ion plasma with RKFT

We will compare our analytical RKFT results to an MD simulation described in Section 4.1. The system parameters that enter into our RKFT calculations and the MD simulation are identical and given in Section 4.2. With the expressions for the two-point cumulants provided in Section 3.3 and one-loop contributions from Appendix B, we compute in Section 4.3 the radial distribution function (RDF),

$$\text{RDF} = \xi(r) + 1 \,, \tag{63}$$

which is directly related to the two-point correlation function $\xi(r)$ for the system described in Section 3, where an ultracold ion plasma is produced from an ultracold many-body system consisting of neutral Rydberg atoms. We then use the computed RDFs to estimate the amount of disorder-induced heating in the ionic system in Section 4.4.

In Section 4.5 we demonstrate the effect of shot noise on the evolution of structure in a system of discrete particles and discuss consequences thereof for any theoretical description of such systems.

## 4.1 Numerical simulation of the ultracold ion plasma

We set up a molecular dynamics (MD) simulation as a reference model to assess the validity of the RKFT results. The simulation propagates $N = 32000$ particles in a box of volume $V = (200\,\mu\text{m})^3$ with periodic boundary conditions following the Hamiltonian equations of motion (46) and (47). This results in a number density of $\bar{\rho} = 4 \cdot 10^9 \text{cm}^{-3}$. The particle mass is assumed to correspond to that of $^{87}$Rb. The initial velocity dispersion of the particles is set by an initial temperature of $T = 100\,\mu K$ to appropriately capture the conditions in an ultracold gas. The amplitude of the interaction potential is set to $A = 10.6895\,k_B T$ corresponding to typical values found for $C_6$ for $^{87}$Rb. The values are chosen such as to approximate conditions typically realised in experiments [2].

The initial anti-correlation effect due to the Rydberg blockade is realised by rejection sampling: Initial particle positions are drawn from a uniform random distribution, but rejected if they fall within the blockade radius $R_b$ around any previously placed particle. The sampling stops once $N$ valid particle positions are drawn. We set the Rydberg blockade radius to $R_b = 5\mu\text{m}$. We create 120 realisations of the system which are then averaged over in order to obtain statistics on the system and thus make it comparable to our RKFT results. In order to reduce computational cost, we introduce a cut-off into the computation of particle interactions which limits the radius within which particle-particle interactions are evaluated. We adjust the cut-off scale for each choice of the Gaussian potential width (52) such that the cut-off radius is enforced when the strength of the interaction potential has decreased to $\sim 0.03\%$ of its maximum value. We find that this is sufficient to fully capture the dynamics of the system.

We do not simulate the dynamics of ground state atoms or electrons. All particles in our simulation either represent neutral Rydberg atoms at initial time $t_\text{i}$ or ions at times $t > t_\text{i}$ (i.e. Rydberg atoms after ionisation).

## 4.2 Choice of system parameters for the ultracold ion plasma

Before we discuss our results we briefly summarise the ingredients and all necessary physical parameters that enter into our RKFT calculations. Identical parameters are used for the MD-

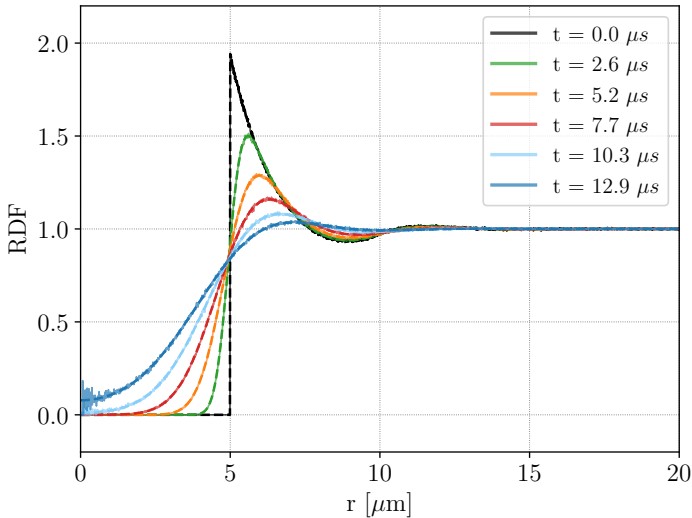

Figure 2: RKFT results (dashed lines) and MD-simulation results (solid lines) for the RDF without particle interactions. Both results overlay perfectly since RKFT provides exact results in this case without the need for a perturbative expansion. The time step between each time-slice presented here corresponds to $\Delta t = 0.1 t_{\text{th}}$ the thermal time scale defined as the ratio between the mean particle distance $d$ and the thermal velocity dispersion $\sigma_{\text{v}} = \frac{\sigma_{\text{p}}}{m}$, $t_{\text{th}} = \frac{d}{\sigma_{\text{v}}}$.

simulations.

For the computation of the two-point density-correlation function (39) we use the equations of motion discussed in Section 3.1. The interaction potential between particle-pairs is given by the Gaussian potential (52). The initial conditions are of the form (53) and the initial power spectrum which characterises the initial distribution of particle positions is sampled directly from the average over initial distributions provided by the MD simulations. This ensures that the starting point of the numerical MD-simulations and the analytical RKFT calculations match as closely as possible to allow a quantitative comparison between analytical and numerical results. To model the conditions in an ultracold gas we set an initial temperature of $T = 100 \, \mu K$. The temperature enters the calculations in terms of the initial momentum self-correlations (i. e. the initial velocity dispersion) of the particles. The mean density $\bar{\rho}$ for the RKFT calculations is computed from $\bar{\rho} = \frac{N}{V}$ where the number of particles $N$, the volume $V$ and the particle mass $m$ are the same as for the MD-simulation (see Section 4.1).

We vary the width of the Gaussian interaction potential $\sigma$ in (52), choosing values from the set $\sigma = \{0.5, 2.5, 5.0, 7.5\}\mu$m. This selection of values already allows us to constrain the regime where RKFT provides the best results and identify its limits.

## 4.3  Results for the evolution of structures in an ultracold ion plasma

In a first step, we perform a consistency check and consider a system without particle-particle interactions. In this case, we can compute the exact result for the correlation function from the free two-point density cumulant $G_{\rho\rho}^{(0)}$ given in (62) without the need for a perturbative expansion. The result is shown in Figure 2 together with the result obtained from numerical MD simulations. As expected, the agreement between our analytical predictions from RKFT and the numerical

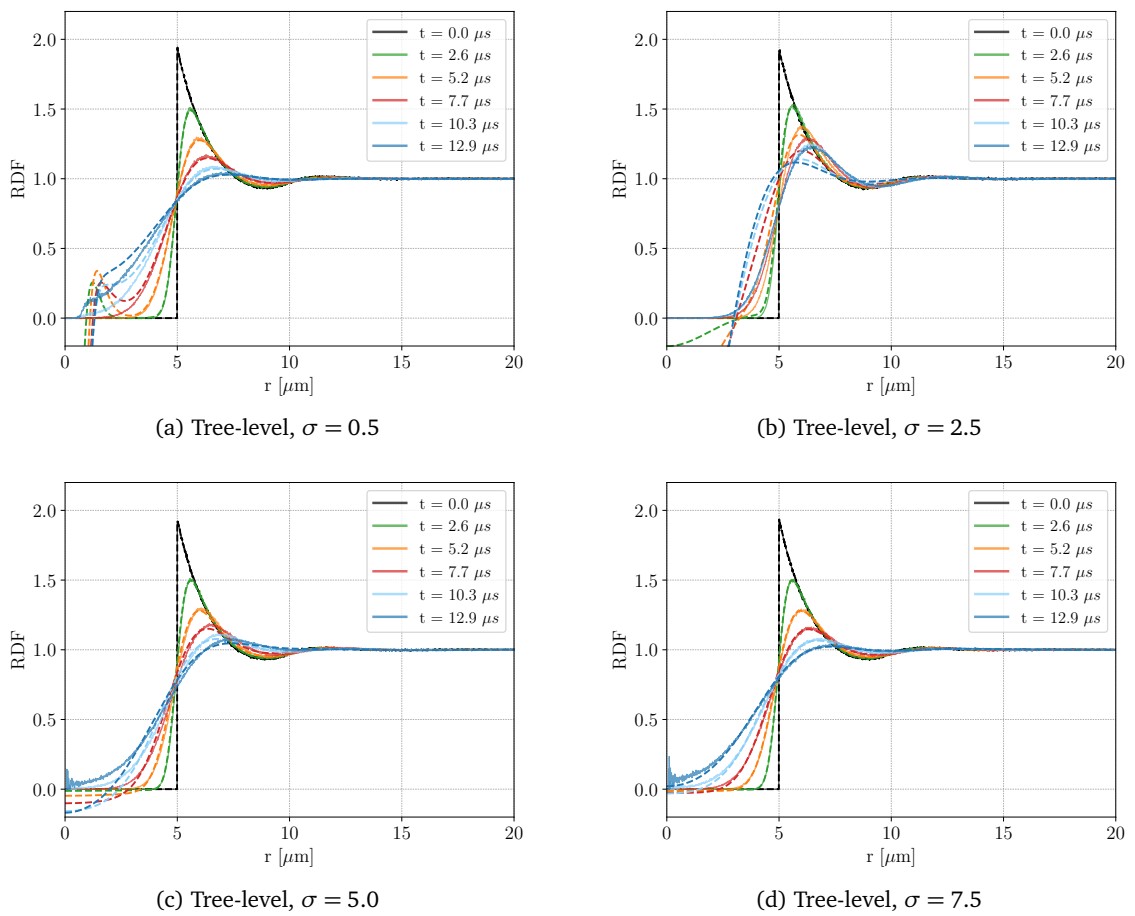

(a) Tree-level, $\sigma = 0.5$

(b) Tree-level, $\sigma = 2.5$

(c) Tree-level, $\sigma = 5.0$

(d) Tree-level, $\sigma = 7.5$

Figure 3: Tree-level RKFT results (dashed lines) and MD-simulation results (solid lines) for the RDF for different potential widths $\sigma$ of a gaussian-shaped potential. The agreement between analytic and simulation results increases with increasing range of the interaction potential between particles.

radial distribution functions at any time in the evolution is exact.

For the interacting system described in Section 3 and 4.2, the analytical tree-level RKFT results (dashed lines) are presented together with the results from the MD-simulations (solid lines) for different choices of the interaction width $\sigma$ in Figure 3. In this case, it is clear that the analytical prediction will not match the MD-simulation results exactly since we neglect loop-corrections to the propagator in (39). We make two observations from Figure 3: (1) It is evident that the agreement between the analytical RKFT predictions and the MD-simulation results improves with increasing potential width. (2) The radial distribution function becomes negative for small radii. The latter behaviour is un-physical since the RDF is, by definition, a non-negative function. The negative values for the RDFs must, therefore, be a consequence of the perturbative approach used in RKFT. This effect becomes worse for decreasing $\sigma$. Both of the above observations can be understood by considering that the Gaussian interaction potential in (52) becomes steeper with decreasing width, which leads to a stronger deflection from inertial particle trajectories due to interactions. As previously discussed at the end of Section 2.1, the deflection from inertial

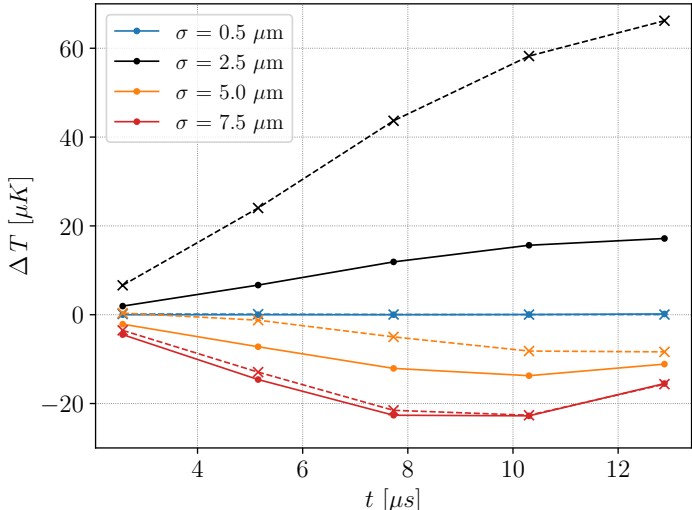

Figure 4: Correlation heating (cooling) temperatures for MD-simulation results (dots) and tree-level RKFT results (crosses) where the RDF has been set to zero for all radii where it falls below zero. This improves the overall agreement between the analytic and simulation predictions. The lines are intended to guide the eye from time slice to time slice at which the correlation temperatures were evaluated.

trajectories is exactly the underlying quantity in which we perturb in RKFT. Thus, higher-order self-energy contributions (i. e. loop-corrections) must become more important on scales where the potential starts to dominate the evolution of the system. In order to quantify and analyse the effect of higher-order contributions, we present the results for the next order in the perturbation series, i. e. the one-loop corrections given by (69) and (70), in Appendix B. We conclude, however, that at one-loop level the RDFs can still become negative and the overall benefit for the accuracy of the RDFs is very small. It appears that the perturbation series converges very slowly (see Appendix B.2) and we will require a replacement for the perturbative ansatz in order to extend the validity of the theory to larger time scales, smaller radii and steeper interaction potentials. We discuss possible approaches in Section 5.

## 4.4 Results for disorder-induced heating using RKFT

Using the results for the RDFs from the previous section, we could now compute the temperature increase due to disorder-induced (or correlation) heating given by (4). However, following the discussion in Section 4.3, we have to account for the fact that the RDFs obtained with RKFT can acquire negative values on small scales which does not have a physical interpretation. Since MD-simulation results show that in this regime the RDFs approach zero, we introduce a cut-off for the RDFs obtained with RKFT setting the values of the RDFs to zero when they become negative.

Our results in Figure 4 show that, using this scheme, we reproduce the trend for the heating temperature correctly. For completeness, we also present and discuss results for disorder-induced heating without introducing the cut-off at small radii of the RDFs in Appendix C. The values for $\Delta T$ obtained with RKFT in Figure 4 appear to be systematically higher than those obtained through MD simulations. The cause is yet unclear and could well be a consequence of missing loop-corrections. For $\sigma = 5\mu$m and $\sigma = 7.5\mu$m we observe a decrease in temperature which

indicates correlation-cooling. Correlation-cooling has been first discussed in [23–25]. It describes the reverse effect to disorder-induced heating, where the temperature decreases after ionisation. In [23–25] correlation-cooling is achieved by either starting from a highly ordered state of the neutral system which then relaxes into a less ordered state after ionisation, or by a sudden change in the interaction potential (e. g. by suddenly decreasing the Debye screening length in a plasma). In our case, however, certain combinations of initial conditions and interaction potential lead to correlation-cooling. We start out from a highly ordered system of neutral Rydberg atoms which are then instantaneously ionised. The strong (anti-)correlations are thus imprinted on the ionised system. For Gaussian potential widths $\sigma$ smaller than the Rydberg blockade radius $R_b$ we observe correlation-heating, whereas for $\sigma \geq R_b$ we achieve correlation-cooling. This suggests that the interplay between strong initial correlations and a long-ranged potential allows the system to relax into a less ordered state, preventing or even reversing correlation heating.

## 4.5 Importance of shot-noise effects

In Section 3.3 we have already noted the appearance of shot-noise terms in our analytical, particle-based RKFT formalism. Shot noise contributions arise due to the discrete nature of the density field, since it is sampled by discrete particles. All terms appearing in the free cumulants (32) which scale as $\bar{\rho}^l$ with a power $l$ less than the number $n_\rho$ of fields $\rho$ are shot-noise contributions. In the example of (62), $n_\rho = 2$ and therefore all terms which scale as $\bar{\rho}^l$ with $l < 2$ are shot-noise terms. In the thermodynamic limit $N \to \infty$ these shot-noise contributions would be negligible compared to the dominant term that scales with $\bar{\rho}^{n_\rho}$. As we will now see, in our case we cannot assume the thermodynamic limit and therefore need to take shot-noise contributions into account as they play an increasingly significant role with decreasing scale.

In Figure 5 we show RDFs computed with RKFT for $\sigma = 5.0\mu m$, where we have neglected all shot-noise contributions. When we compare Figure 5 to Figure 3c, where we include shot noise contributions, we see that shot-noise considerably alters the RDFs even on large scales. Taking into account shot-noise contributions, and thereby accounting for the fact that the system is actually composed of discrete particles, dramatically increases the agreement of our analytical RKFT results with simulations. The same effect can be observed for any value of $\sigma$. These results suggest that any theoretical description of an ensemble of particles where the thermodynamic limit cannot be applied must include shot noise contributions in order to realistically predict the evolution of the many-body system. While shot-noise contributions arise naturally in (R)KFT due to its particle-based formulation, methods based on the Vlasov or hydrodynamical equations do not account for shot-noise effects because they describe the system in terms of continuous fields [26].

# 5 Conclusions

In this work we have shown how the (R)KFT formalism, initially intended to study cosmic structure formation, can be used to study the evolution of an ultra-cold, initially (anti-)correlated plasma produced from a Rydberg-blockaded neutral gas. The RKFT perturbation scheme developed in [15] has allowed us to resum an infinite subset of perturbative contributions which only contain free two-point cumulants but involve arbitrary many particle-particle interactions. This corresponds to a resummation of all contributions that do not lead to mode-coupling caused by the interaction potential. It is the tree-level result for the two-point density cumulant. Loop-corrections to the tree-level result can then be obtained by taking into account vertex-terms which

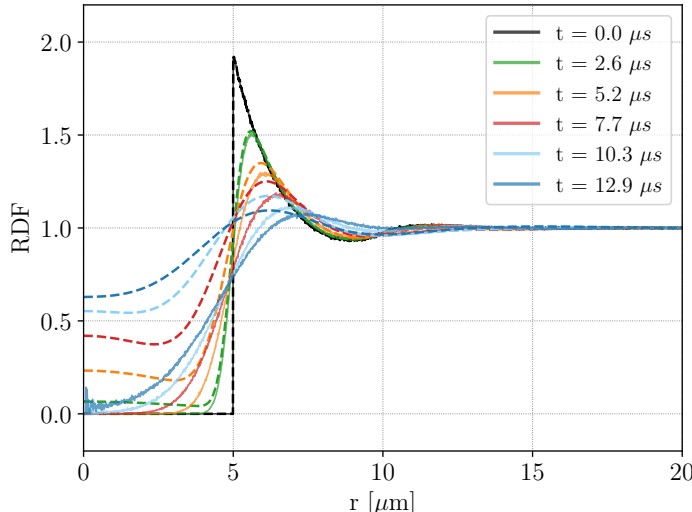

Figure 5: RKFT results when neglecting shot noise contributions (dashed lines) and MD-simulation results (solid lines) for the RDF for a potential width $\sigma = 5\mu$m. Neglecting shot-noise contributions has a strong effect, especially on small scales, and leads to a large deviation from numerical MD-simulation results.

introduce mode-coupling effects due to interactions.

We have computed the tree-level RDF for different evolution times which we then used to determine the amount of disorder-induced heating that would be produced after ionisation of such a Rydberg-blockaded gas. For simplicity, we have assumed a Gaussian-shaped interaction potential between particle pairs. By varying the width of the Gaussian potential, we have studied how well our formalism captures the non-linear dynamics of the evolving system. We have shown that for more shallow, long-ranged potentials our results for the RDFs as well as the disorder-induced heating computed from them agree remarkably well with MD simulations. Even more interestingly, we have seen that for long ranged-potentials the effect of correlation-heating is, in fact, reversed and we observe correlation-cooling instead. In our case, correlation-cooling occurs as a result of the strong initial (anti-)correlations in the neutral Rydberg gas prior to ionisation in combination with the long range of the interaction potential. This implies that the order in the system due to initial (anti-)correlations is greater than the order imposed by the interaction potential after ionisation. We have furthermore demonstrated that shot-noise effects which arise due to the discrete nature of a many-particle system yield a significant contribution to the RDF and must be taken into account. Typically, such effects are neglected when approaches based on the Vlasov or hydrodynamic equations are used since they describe the system in terms of continuous fields.

The low computational cost of the RKFT computations makes it an ideal tool to scan large parameter spaces, varying input parameters such as the particle interaction potential, initial correlations, temperature or density. It can be used to find optimal parameters for which disorder-induced heating during plasma production is minimised (or even reversed) and strong coupling parameters can be achieved.

For steep, short-ranged potentials, however, our results deviate from MD-simulations. Especially on small scales the accuracy of our predicted RDFs suffers severely and can even produce negative values for the RDFs which is ruled out by definition. We have argued, that this is due

to the truncation of our perturbative series at zeroth (tree-level) order and that higher order contributions (i.e. loop-corrections) should improve our results. In a first attempt to alleviate this problem we have computed the one-loop corrections to the correlation function. However, this did not yield any improvement for our results and we believe the reason for this is the following: The dressed free cumulants (32) appearing in the RKFT action (31) characterise both the initial statistics and the dynamics of the system under consideration. Specifically, a cumulant $G^{(0)}_{f \cdots f \mathcal{F} \cdots \mathcal{F}}$ of $n_f$ phase-space density fields $f$ and $n_{\mathcal{F}}$ dressed response fields $\mathcal{F}$ describes the $n_{\mathcal{F}}$-th order response of the free $n_f$-point density cumulant to particle-particle interactions. The tree-level result $\Delta_{ff}$ of the two-point density cumulant $G_{ff}$ resums an infinite subset of perturbative contributions involving arbitrary powers of the linear response cumulant $G^{(0)}_{f\mathcal{F}}$. This can be seen by expanding the functional inverse (45) in powers of $G^{(0)}_{f\mathcal{F}}$. Loop-corrections describe perturbative contributions also involving nonlinear response cumulants, $n_{\mathcal{F}} \geq 2$, via the vertices. However, each finite loop-order correction only involves a finite number of these nonlinear response cumulants which might introduce an inconsistency with the infinitely resummed linear response cumulants, leading to the poor convergence behaviour of the loop expansion. Should that be the case, an additional 1PI/Dyson resummation of the one-loop self energy contributions might mend this inconsistency. Alternatively, one could apply non-perturbative techniques, such as the functional renormalisation group [27, 28], to overcome convergence issues of perturbative expansions entirely. We will investigate these approaches in future work.

In this first application of the RKFT framework to ultracold correlated systems we have made a few simplifying assumptions such as modelling the interaction potential as a Gaussian and neglecting electrons as a separate particle species in the ultracold plasma stage. The Gaussian potential can be easily replaced by any other potential, e.g. a Yukawa potential which includes Debye-screening by the electrons. This does not require any changes in the theoretical framework.[2] It is also easily possible to set different interaction potentials for the neutral Rydberg gas and the ionised plasma for the computation of disorder-induced heating. A full treatment of the ionised plasma including electrons, however, requires to extend our theory in order to describe a two-component system treating electrons and ions as different particle species. Our theoretical framework does allow such an extension. It has been previously demonstrated in the cosmological setting where a two-component system consisting of gas and dark-matter particles has been studied. The crucial part of such an extension for the plasma system is then the modelling of the complex interactions between electrons and ions. This is a highly non-trivial task and will have to be explored in future work.

## Acknowledgements

The authors would like to thank Tristan Daus, Markus Oberthaler, Asier Piñeiro Orioli, Christophe Pixius, Manfred Salmhofer, Bjoern Malte Schaefer and Veit Stooß for their valuable input and fruitful discussions.

**Funding information**   This work was supported in part by the Heidelberg Center for Quantum Dynamics and the the Heidelberg Graduate School of Fundamental Physics (HGSFP) funded by the Deutsche Forschungsgemeinschaft (DFG, German Research Foundation)

---

[2]Our reasons for choosing a Gaussian shaped potentials were detailed in Sec. 3.1

– 24838972. EK is funded by the Deutsche Forschungsgemeinschaft (DFG, German Research Foundation) – 452923686. ASc acknowledges funding by the DOE Office of Science, Office of Advanced Scientific Computing Research (ASCR) Quantum Computing Application Teams program, under fieldwork proposal number ERKJ347. ASa acknowledges funding by the Deutsche Forschungsgemeinschaft (DFG, German Research Foundation, Project-ID 273811115, SFB 1225 ISOQUANT) and DFG Priority Program "GiRyd 1929" (Grant No. DFG WE2661/12-1).

## A  Two-point cumulant in the case of zero initial temperature

In the case where all initial momentum correlations – including momentum auto-correlations (i.e. initial temperature $T_i = 0$) – are neglected, we can give an analytical expression for the tree-level result of the two-point density cumulant $G_{\rho\rho}(1,2)$ in (39).

To determine the macroscopic propagator, we first need to insert the expression (58) for $G^{(0)}_{\rho\mathcal{F}}$ into the functional inverse (45) defining the causal propagators $\Delta_R$ and $\Delta_A$. This inverse can be calculated fully analytically by means of Laplace transforms where the fact that $G^{(0)}_{\rho\mathcal{F}}(1,2)$ is time-translation invariant, i.e. it only depends on $t_1$ and $t_2$ in terms of their difference, is exploited. The result reads

$$\Delta_R(1,2) = \Delta_A(2,1) = \mathcal{I}(1,2) - (2\pi)^3 \delta_D(\vec{k}_1 + \vec{k}_2) c(k_1) k_1 \sin(k_1(t_1 - t_2) c(k_1)) \Theta(t_1 - t_2). \quad (64)$$

The function $c(k_1)$ appearing here is defined as

$$c(k_1) := \sqrt{\frac{\bar{\rho} v(k_1)}{m}}. \quad (65)$$

Inserting the results for $\Delta_R$ and $\Delta_A$ as well as the expression (62) for $G^{(0)}_{\rho\rho}$ into the $ff$-component of (44) then yields the density-density propagator,

$$\Delta_{\rho\rho}(1,2) = (2\pi)^3 \delta_D(\vec{k}_1 + \vec{k}_2)(\bar{\rho} + \bar{\rho}^2 P^{(i)}_\delta(k_1)) \cos(k_1 t_1 c(k_1)) \cos(k_2 t_2 c(k_2)), \quad (66)$$

which corresponds to the tree-level result for the two-point density cumulant $G_{\rho\rho}(1,2)$ according to (39).

## B  One-loop contributions

In order to compute the self-energy contributions in (39), we start by defining an *exact macroscopic propagator G* as the matrix of all fully interacting macroscopic two-point cumulants,

$$G(1,2) := \begin{pmatrix} G_{ff} & G_{f\beta} \\ G_{\beta f} & G_{\beta\beta} \end{pmatrix}(1,2). \quad (67)$$

To organise the different loop contributions to $G$, we follow the conventional approach from quantum and statistical field theory and introduce the *macroscopic self-energy* $\Sigma$, defined via

$$G(1,2) =: \Delta(1,2) + (\Delta \cdot \Sigma \cdot \Delta)(1,2). \quad (68)$$

It corresponds to the sum of all 1PI two-point loop contributions with both their external tree-level propagators stripped off and thus acts like an effective two-point vertex. As such it inherits the

causal structure of the tree-level vertices, meaning that $\Sigma_{\beta\beta}$ is symmetric in its time arguments, $\Sigma_{\beta f}$ and $\Sigma_{f\beta}$ have a retarded and an advanced time-dependence, respectively, and $\Sigma_{ff}$ vanishes identically.

In order to avoid redundancies when computing the propagator corrections to some given loop order, it is advantageous to first compute the respective loop contributions to the self-energies, and afterwards use

$$
\begin{aligned}
G_{ff}(1,2) = \Delta_{ff}(1,2) &+ \left(\Delta_{f\beta} \cdot \Sigma_{\beta\beta} \cdot \Delta_{\beta f}\right)(1,2) \\
&+ \left(\Delta_{f\beta} \cdot \Sigma_{\beta f} \cdot \Delta_{ff}\right)(1,2) \\
&+ \left(\Delta_{ff} \cdot \Sigma_{f\beta} \cdot \Delta_{\beta f}\right)(1,2),
\end{aligned}
\tag{69}
$$

$$
G_{f\beta}(1,2) = G_{\beta f}(2,1) = \Delta_{f\beta}(1,2) + \left(\Delta_{f\beta} \cdot \Sigma_{\beta f} \cdot \Delta_{f\beta}\right)(1,2),
\tag{70}
$$

to obtain the resulting expressions for the propagators. In [15] a diagrammatic language was introduced to represent the propagators and vertices of RKFT. Using this one can conveniently express the one-loop contributions in terms of appropriate Feynman diagrams. In this work we are only interested in the two-point spatial density cumulant $G_{\rho\rho}$. The one-loop integral expressions for the corresponding spatial density self-energies appearing in (69) and (70) are provided in Appendix B.1.

## B.1 Expressions for the self-energies at one-loop order

$$
\begin{aligned}
\Sigma_{\beta\beta}^{(1\text{-loop})}(1,2) = \ &\frac{1}{2}\int \mathrm{d}3'\mathrm{d}4'\mathrm{d}\bar{3}\,\mathrm{d}\bar{4}\,\mathcal{V}_{\beta\rho\rho}(1,3',4')\mathcal{V}_{\beta\rho\rho}(2,\bar{3},\bar{4})\Delta_{\rho\rho}(-3',-\bar{3})\Delta_{\rho\rho}(-4',-\bar{4}) \\[4pt]
&+\int \mathrm{d}3'\mathrm{d}4'\mathrm{d}\bar{3}\,\mathrm{d}\bar{4}\,\mathcal{V}_{\beta\rho\rho}(1,3',4')\mathcal{V}_{\beta\rho\beta}(2,\bar{3},\bar{4})\Delta_{\rho\rho}(-3',-\bar{3})\Delta_{\rho\beta}(-4',-\bar{4}) \\[4pt]
&+\int \mathrm{d}3'\mathrm{d}4'\mathrm{d}\bar{3}\,\mathrm{d}\bar{4}\,\mathcal{V}_{\beta\rho\beta}(1,3',4')\mathcal{V}_{\beta\rho\rho}(2,\bar{3},\bar{4})\Delta_{\rho\rho}(-3',-\bar{3})\Delta_{\beta\rho}(-4',-\bar{4}) \\[4pt]
&+\int \mathrm{d}3'\mathrm{d}4'\mathrm{d}\bar{3}\,\mathrm{d}\bar{4}\,\mathcal{V}_{\beta\rho\beta}(1,3',4')\mathcal{V}_{\beta\beta\rho}(2,\bar{3},\bar{4})\Delta_{\rho\beta}(-3',-\bar{3})\Delta_{\beta\rho}(-4',-\bar{4}) \\[4pt]
&+\frac{1}{2}\int \mathrm{d}3'\mathrm{d}4'\mathrm{d}\bar{3}\,\mathrm{d}\bar{4}\,\mathcal{V}_{\beta\rho\rho}(1,3',4')\mathcal{V}_{\beta\beta\beta}(2,\bar{3},\bar{4})\Delta_{\rho\beta}(-3',-\bar{3})\Delta_{\rho\beta}(-4',-\bar{4}) \\[4pt]
&+\frac{1}{2}\int \mathrm{d}3'\mathrm{d}4'\mathrm{d}\bar{3}\,\mathrm{d}\bar{4}\,\mathcal{V}_{\beta\beta\beta}(1,3',4')\mathcal{V}_{\beta\rho\rho}(2,\bar{3},\bar{4})\Delta_{\beta\rho}(-3',-\bar{3})\Delta_{\beta\rho}(-4',-\bar{4}) \\[4pt]
&+\frac{1}{2}\int \mathrm{d}3'\mathrm{d}\bar{3}\,\mathcal{V}_{\beta\beta\rho\rho}(1,2,3',\bar{3})\Delta_{\rho\rho}(-3',-\bar{3}) \\[4pt]
&+\int \mathrm{d}3'\mathrm{d}\bar{3}\,\mathcal{V}_{\beta\beta\rho\beta}(1,2,3',\bar{3})\Delta_{\rho\beta}(-3',-\bar{3})
\end{aligned}
\tag{71}
$$

$$\Sigma_{\beta\rho}^{(1\text{-loop})}(1,2) = \int d3' d4' d\bar{3}\, d\bar{4}\, \mathcal{V}_{\beta\rho\rho}(1,3',4')\, \mathcal{V}_{\rho\rho\beta}(2,\bar{3},\bar{4})\, \Delta_{\rho\rho}(-3',-\bar{3})\, \Delta_{\rho\beta}(-4',-\bar{4})$$

$$+\frac{1}{2}\int d3' d4' d\bar{3}\, d\bar{4}\, \mathcal{V}_{\beta\rho\rho}(1,3',4')\, \mathcal{V}_{\rho\beta\beta}(2,\bar{3},\bar{4})\, \Delta_{\rho\beta}(-3',-\bar{3})\, \Delta_{\rho\beta}(-4',-\bar{4})$$

$$+\int d3' d4' d\bar{3}\, d\bar{4}\, \mathcal{V}_{\beta\rho\beta}(1,3',4')\, \mathcal{V}_{\rho\beta\rho}(2,\bar{3},\bar{4})\, \Delta_{\rho\beta}(-3',-\bar{3})\, \Delta_{\beta\rho}(-4',-\bar{4})$$

$$+\frac{1}{2}\int d3' d\bar{3}\, \mathcal{V}_{\beta\rho\rho\rho}(1,2,3',\bar{3})\, \Delta_{\rho\rho}(-3',-\bar{3})$$

$$+\int d3' d\bar{3}\, \mathcal{V}_{\beta\rho\rho\beta}(1,2,3',\bar{3})\, \Delta_{\rho\beta}(-3',-\bar{3}) \tag{72}$$

In the following we give the expressions for the free cumulants (see [14] for details) required for the calculation of the one-loop contributions above. In order to keep the expressions compact we introduce the following abbreviations for the response factor

$$b(1,2) = \vec{k}_1 \cdot \vec{k}_2 \,\frac{t_1 - t_2}{m}\, \Theta(t_1 - t_2), \tag{73}$$

and the damping factor

$$D(1,\ldots,,n) = \exp\left(-\frac{k_B T_i}{2m}\left(\vec{k}_1 t_1 + \cdots + \vec{k}_n t_n\right)^2\right). \tag{74}$$

$$G_{\rho\rho\rho}^{(0)}(1,2,3) = (2\pi)^3 \delta_D\left(\vec{k}_1 + \vec{k}_2 + \vec{k}_3\right)$$
$$\times \Bigg[\bar{\rho}\, D(1,2,3)$$
$$+ \bar{\rho}^2 P_\delta^{(i)}(k_1)\, D(1)\, D(2,3) + \bar{\rho}^2 P_\delta^{(i)}(k_2)\, D(2)\, D(1,3)$$
$$+ \bar{\rho}^2 P_\delta^{(i)}(k_3)\, D(3)\, D(1,2)\Bigg], \tag{75}$$

$$G_{\rho\rho\mathcal{F}}^{(0)}(1,2,3) = -i(2\pi)^3 \delta_D\left(\vec{k}_1 + \vec{k}_2 + \vec{k}_3\right) v(k_3)$$
$$\times \Bigg[\bar{\rho}\left(b(1,3) + b(2,3)\right) D(1,2,3)$$
$$+ \bar{\rho}^2 P_\delta^{(i)}(k_1)\, b(2,3)\, D(1)\, D(2,3)$$
$$+ \bar{\rho}^2 P_\delta^{(i)}(k_2)\, b(1,3)\, D(2)\, D(1,3)\Bigg], \tag{76}$$

$$G_{\rho\mathcal{F}\mathcal{F}}^{(0)}(1,2,3) = -(2\pi)^3 \delta_D\left(\vec{k}_1 + \vec{k}_2 + \vec{k}_3\right) v(k_2)\, v(k_3)$$
$$\times \bar{\rho}\left(b(1,2) + b(3,2)\right)\left(b(1,3) + b(2,3)\right) D(1,2,3), \tag{77}$$

$$G^{(0)}_{\rho\rho\rho\mathcal{F}}(1,2,3,4) = -i(2\pi)^3 \delta_{\mathrm{D}}\big(\vec{k}_1 + \vec{k}_2 + \vec{k}_3 + \vec{k}_4\big) v(k_4)$$

$$\times \Big[ \bar{\rho}\big(b(1,4) + b(2,4) + b(3,4)\big) D(1,2,3,4)$$

$$+ \bar{\rho}^2 P^{(\mathrm{i})}_\delta(k_1)\big(b(2,4) + b(3,4)\big) D(1) D(2,3,4)$$

$$+ \bar{\rho}^2 P^{(\mathrm{i})}_\delta(k_2)\big(b(1,4) + b(3,4)\big) D(2) D(1,3,4)$$

$$+ \bar{\rho}^2 P^{(\mathrm{i})}_\delta(k_3)\big(b(1,4) + b(2,4)\big) D(3) D(1,2,4)$$

$$+ \bar{\rho}^2 P^{(\mathrm{i})}_\delta(|\vec{k}_1 + \vec{k}_2|)\, b(3,4) D(1,2) D(3,4)$$

$$+ \bar{\rho}^2 P^{(\mathrm{i})}_\delta(|\vec{k}_2 + \vec{k}_3|)\, b(1,4) D(2,3) D(1,4)$$

$$+ \bar{\rho}^2 P^{(\mathrm{i})}_\delta(|\vec{k}_1 + \vec{k}_3|)\, b(2,4) D(1,3) D(2,4) \Big], \tag{78}$$

$$G^{(0)}_{\rho\rho\mathcal{F}\mathcal{F}}(1,2,3,4) = -(2\pi)^3 \delta_{\mathrm{D}}\big(\vec{k}_1 + \vec{k}_2 + \vec{k}_3 + \vec{k}_4\big) v(k_3) v(k_4)$$

$$\times \Big[ \bar{\rho}\big(b(1,3) + b(2,3) + b(4,3)\big)\big(b(1,4) + b(2,4) + b(3,4)\big) D(1,2,3,4)$$

$$+ \bar{\rho}^2 P^{(\mathrm{i})}_\delta(k_1)\big(b(2,3) + b(4,3)\big)\big(b(2,4) + b(3,4)\big) D(1) D(2,3,4)$$

$$+ \bar{\rho}^2 P^{(\mathrm{i})}_\delta(k_2)\big(b(1,3) + b(4,3)\big)\big(b(1,4) + b(3,4)\big) D(2) D(1,3,4)$$

$$+ \bar{\rho}^2 P^{(\mathrm{i})}_\delta(|\vec{k}_1 + \vec{k}_3|)\, b(1,3)\, b(2,4) D(1,3) D(2,4)$$

$$+ \bar{\rho}^2 P^{(\mathrm{i})}_\delta(|\vec{k}_2 + \vec{k}_3|)\, b(2,3)\, b(1,4) D(2,3) D(1,4) \Big], \tag{79}$$

$$G^{(0)}_{\rho\mathcal{F}\mathcal{F}\mathcal{F}}(1,2,3,4) = i(2\pi)^3 \delta_{\mathrm{D}}\big(\vec{k}_1 + \vec{k}_2 + \vec{k}_3 + \vec{k}_4\big) v(k_2) v(k_3) v(k_4)$$

$$\times \bar{\rho}\big(b(1,2) + b(3,2) + b(4,2)\big)\big(b(1,3) + b(2,3) + b(4,3)\big)$$

$$\times \quad \big(b(1,4) + b(2,4) + b(3,4)\big) D(1,2,3,4). \tag{80}$$

## B.2 One-loop corrected results for RDFs

In Figures 6a through 6d we show the results obtained by including the next order in perturbation, i. e. the one-loop corrections given in (69) and (70). We can see that the one-loop contributions add additional power on small scales. This reduces the problem of negative RDFs to some extent, but not on all scales. This suggests that higher-order contributions are necessary to completely avoid negative values for the RDFs on all scales.

However, the overall improvement from one-loop-corrections for the accuracy of our RDFs seems to be small, and in some cases even contrary: For short-ranged interaction potentials with $\sigma = \{0.5, 2.5\}\mu$m, one-loop-corrections improve the agreement between RKFT and MD results by a small amount. For long-ranged potentials $\sigma = \{5.0, 7.5\}\mu$m, on the other hand, the tree-level results shown in Figure 3 agree better with MD-simulations since one-loop corrections add too much power. Since long-ranged potentials affect particle trajectories on larger scales, loop contributions should also contribute on larger scales. If the perturbation series converges, one would expect that the next higher loop-order contributions would cancel any excess power provided by one-loop corrections.

However, the perturbation series seems to converge very slowly – if at all – since the amplitudes of the one-loop contributions appear to be of the same order as the tree-level results. Since

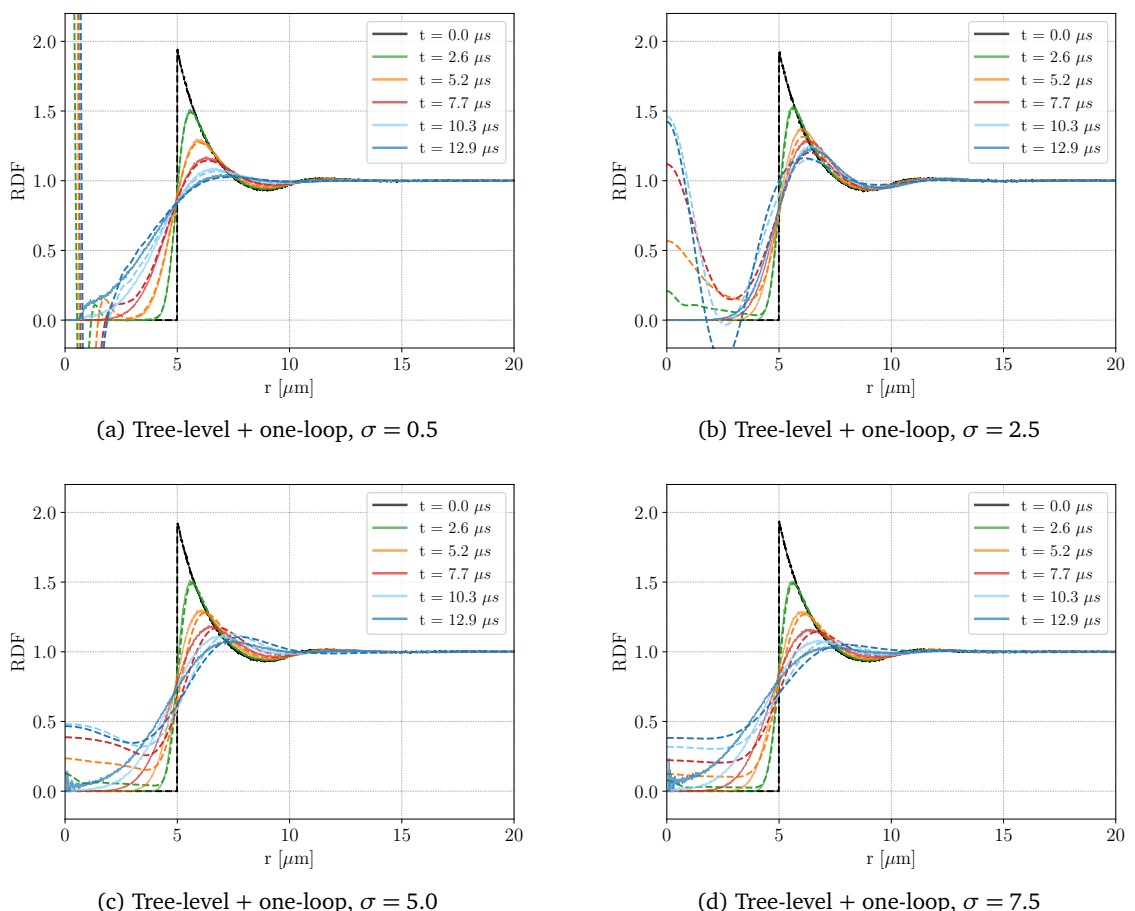

(a) Tree-level + one-loop, $\sigma = 0.5$

(b) Tree-level + one-loop, $\sigma = 2.5$

(c) Tree-level + one-loop, $\sigma = 5.0$

(d) Tree-level + one-loop, $\sigma = 7.5$

Figure 6: Tree-level + one-loop RKFT results (dashed lines) and MD-simulation results (solid lines) for the RDF. We do not observe a consistent improvement over the tree-level RKFT results by including one-loop corrections. In particular for large potential widths, including one-loop corrections even decreases the agreement with the MD-simulation, which may be an indication that a resummation of higher-order corrections is needed.

convergence of the expansion series used so far in RKFT is not guaranteed it will be necessary to find a replacement for the perturbative ansatz in order to extend the validity of the theory to larger time scales, smaller radii and steeper interaction potentials. We discuss possible approaches in Section 5.

## C  Disorder-induced heating using RKFT without cut-off

We see in Figure 7 that the tree-level RKFT results without a cut-off in the RDF consistently under-estimate the heating temperature and even predict heating temperatures with an opposite sign compared to MD simulations. This does not come as a surprise, since we have already seen in Section 4.3 that the RDFs obtained with RKFT become (strongly) negative at small scales, especially for small values of $\sigma$. For large $\sigma$, however, we see that the analytically predicted heating

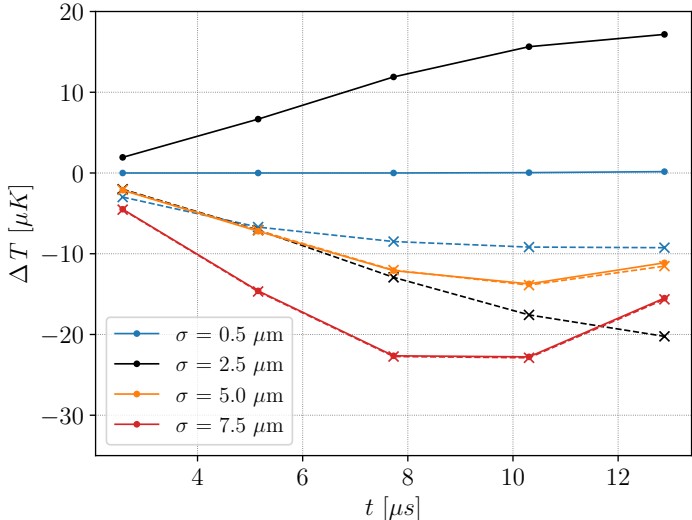

Figure 7: Correlation energies for MD (dots) and tree-level RKFT results (crosses) given by (4) without introducing a cut-off for the RDFs. The lines are intended to guide the eye from time-slice to time-slice at which the correlation energies were evaluated.

temperatures without a cut-off agree very well with those obtained from simulations.

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
