# Peer review of "Ultracold plasmas from strongly anti-correlated Rydberg gases in the Kinetic Field Theory formalism"

_SciPost Physics_

## Round 1 · Referee Report · Anonymous · 2023-6-10

Strengths

1) mostly clear writing
2) a quite self contained introduction to RKFT
3) a thorough comparison of analytics with numerics, both quite challenging
4) an interesting interdisciplinary adaptation of potentially very powerful methods

Weaknesses

1) The use of a seemingly unrealistic interaction potential is insufficiently strongly highlighted at the outset.
2) Insufficient discussion of where the (semi-)analytical method is expected to be superior to MD simulations

Report

The present theoretical manuscript describes an application of resummed kinetic field theory (RKFT) to ultracold Plasmas, comparing analytical results with molecular dynamics simulations.

Resummed kinetic field theory was recently developed by some of the authors in the context of cosmological structure formation, and represents an interesting adaptation of path integral concepts from quantum field theory to classical mechanics, by essentially constraining the paths integrated over to the classical ones. It then appears to enable similar perturbative expansions of correlation functions as the better known quantum framework.

The ultracold plasma in question arises when a strongly blockaded assembly of cold Rydberg atoms is suddenly ionised, a process that is actively investigated experimentally. A key question that motivates the present work, is the subsequent increase or decrease of the plasma temperature due to conversion of initial state correlations into kinetic energy.

The present work is an interesting interdisciplinary venture that tackles a challenging problem and convinces regarding validity through the direct comparison of a conceptually highly involved theory (semi-analytical methods) with similarly involved numerical simulations. The level of agreement found seems to rule out any major practical or conceptual errors, while the deviations found then allow the authors to discuss the limitation of the analytical method, to propose possible remedies for the future.

Despite the heavy formalism I felt the authors have mostly described the procedure well and accessible, this still can be improved in several places as listed below. More importantly, I think two items that should be discussed more thoroughly to enable the reader to assess the importance of the manuscript for their own work are the following:

Instead of an actual screened Coulomb potential, the authors use the repulsive Gaussian in Eq. (52). This appears to be a major change of the physics, and I thus not sure to what extent it is valid in abstract, introduction and conclusion to claim that an “ultracold plasma” is treated here. I think the results remain interesting, just that some qualifiers such as “simplified model potential” should be scattered more liberally though the above sections of the manuscript. It should also be discussed more whether / how the authors have chosen the parameters A and sigma of this potential to make the physics more closely resembling to that of the true potential
(B) It is not really made clear for which sort of problems the present analytical method will represent an advantage over the MD simulation. After all, through the direct comparison, the authors demonstrate that their present study could directly be done numerically. Presumably that is not true for all possible scenarios, and these should then be listed somewhere.

I think the points above and below should be discussed more clearly, but with those changes the manuscript would then meet publication criteria.

Requested changes

1) Points (A) and (B) above should be addressed in the discussion.
2) Sentences are often not as clear as they can be, the most prominent example is the last sentence on page 3 “In this work we propose a….approach”, where really the proposal here is only to apply this approach to an u-cold plasma. Also the abstract contains a few comparisons, without comparison object (…better insight than what….?)
3) Too many variables or parameters are not, or not clearly, defined. (In Eq. (1), above (2), correlation function in (3),
4) It would be better to give some preview statement regarding the utility of the generating function Z somewhere near its first occurrence, Eq. (14). Right now, one has to wait until Eq. (23) to finally see what those are good for.
5) What is meant by “Treat the Rydberg atoms as hard sphere” is quite unclear on page 11, and then only becomes clear on page 12.
6) The last sentence of section 3.1. appears to suggest that a Gaussian potential such as (52) between ions can be realised. I am not aware of that and did find no support in [2].
7) I am confused why there are non-vanishing correlations outside the blockade radius at t=0 in Fig. 2. , this should be discussed.
8) The sentence “This corresponds to a resummation ….” in the conclusion is quite unclear.
9) For Fig. 6 I would find it more useful to compare tree level and one loop results for RKFT, instead of RKFT with MD. Maybe this could at least be an extra figure.

---

## Editorial Decision

awaiting_resubmission